# Impact of Misdiagnosis in Case-Control Studies of Myalgic Encephalomyelitis/Chronic Fatigue Syndrome

**DOI:** 10.3390/diagnostics13030531

**Published:** 2023-02-01

**Authors:** João Malato, Luís Graça, Nuno Sepúlveda

**Affiliations:** 1Instituto de Medicina Molecular João Lobo Antunes, Faculdade de Medicina, Universidade de Lisboa, 1649-028 Lisboa, Portugal; 2CEAUL—Centro de Estatística e Aplicações da Universidade de Lisboa, 1749-016 Lisboa, Portugal; 3Faculty of Mathematics and Information Science, Warsaw University of Technology, 00-662 Warszawa, Poland

**Keywords:** misdiagnosis, misclassification, association studies, simulation, statistical power, ME/CFS

## Abstract

Misdiagnosis of myalgic encephalomyelitis/chronic fatigue syndrome (ME/CFS) can occur when different case definitions are used by clinicians (relative misdiagnosis) or when failing the genuine diagnosis of another disease (misdiagnosis in a strict sense). This problem translates to a recurrent difficulty in reproducing research findings. To tackle this problem, we simulated data from case-control studies under misdiagnosis in a strict sense. We then estimated the power to detect a genuine association between a potential causal factor and ME/CFS. A minimum power of 80% was obtained for studies with more than 500 individuals per study group. When the simulation study was extended to the situation where the potential causal factor could not be determined perfectly (e.g., seropositive/seronegative in serological association studies), the minimum power of 80% could only be achieved in studies with more than 1000 individuals per group. In conclusion, current ME/CFS studies have suboptimal power under the assumption of misdiagnosis. This power can be improved by increasing the overall sample size using multi-centric studies, reporting the excluded illnesses and their exclusion criteria, or focusing on a homogeneous cohort of ME/CFS patients with a specific pathological mechanism where the chance of misdiagnosis is reduced.

## 1. Introduction

Myalgic encephalomyelitis/chronic fatigue syndrome (ME/CFS) is a heterogeneous disease whose hallmark symptom is unexplained persistent fatigue [1] or post-exertional malaise upon minimal physical or mental effort [2]. Disease heterogeneity derives from the coexistence of multiple pathological mechanisms in the same patient. Examples of these mechanisms are leaky gut [3], the presence of deleterious autoantibodies [4], oxidative stress [5,6], persisting viral infections [7,8], and severe longstanding stress [9]. Unsurprisingly, research efforts to find a biomarker for disease diagnosis have failed over the years.

Current diagnosis of ME/CFS is performed via multiple polythetic disease definitions where some but not all core symptoms should be present in a suspected case [10]. A differential diagnosis should also be made by excluding known diseases that could explain fatigue and other major symptoms (e.g., multiple sclerosis and diabetes). Given the multiplicity of existing disease definitions, it is possible to diagnose a suspected case of ME/CFS by a consensual case definition but not by an alternative one [11]. This situation is here referred to as relative misdiagnosis because it is only admissible when considering the outcome of a given case definition relative to the one from another case definition. This type of misdiagnosis is typically present when comparing or combining data from studies using different case definitions. Given that consensual definitions for ME/CFS are both difficult to find and suboptimal to patient/control discrimination [12], some efforts were made to investigate empirical approaches to ME/CFS diagnosis [13,14,15].

Misdiagnosis, in a strict sense, arises from the situation where ME/CFS-diagnosed individuals, irrespective of the case definition, are genuine patients of another disease. This has been illustrated in a patient initially diagnosed with ME/CFS but was found to have a rare autosomal adult-onset disorder [16]. This misdiagnosis can result from random fluctuations in the natural, pathological process of the exclusionary disease (e.g., low-graded remitting/relapsing multiple sclerosis). It can also emerge from limited resources to run the battery of tests necessary to exclude all known diseases that could explain fatigue; for example, not performing whole-genome sequencing to exclude rare genetic diseases. There is also ambiguity around the exclusionary criteria themselves, which leaves clinicians unsure of what illnesses should be actually excluded [17]. Therefore, this type of misdiagnosis seems inevitably present in ME/CFS studies [18].

In this paper, we performed a simulation study to determine the statistical power of detecting associations with ME/CFS under misdiagnosis in a strict sense; relative misdiagnosis is beyond the scope of this paper because it is more related to a discussion about the different case definitions, as made elsewhere [19,20]. We also investigated the impact of imperfect sensitivity/specificity for the presence of a given antibody that could be causing ME/CFS. Finally, we extended our analysis to discuss the statistical power of two published studies [21,22].

## 2. Statistical Methodology

### 2.1. Formulation of the Problem

Let us assume a typical case-control study in which diagnosed ME/CFS patients and healthy controls were matched for possible confounding factors, such as age, gender, and body mass index. The main objective of this study is to investigate the association between a candidate causal factor (e.g., a genetic factor or the occurrence of a given infection) and ME/CFS. For simplicity, let us assume that this factor has only two possible values, present and absent; the probabilities for that factor being present in healthy controls and suspected cases are represented by $\theta_{0}$ and $\theta_{1}$, respectively. In general, the respective data are given by a two-way contingency table (Table 1).

Statistically speaking, we aim to investigate the evidence for an association between ME/CFS and the causal factor. This is translated to the following hypotheses
$$H_{0}:\theta_{0}=\theta_{1}\;\mathrm{versus}\;H_{1}:\theta_{0}\neq \theta_{1}\;.$$One can then use the classical Pearson’s $\chi^{2}$ test, where *p*-values < 0.05 indicate a significant association at the 5% significance level.

In this scenario, our objective is to study the impact of misdiagnosis on the power of the Pearson’s $\chi^{2}$ test to detect an association with the disease. With this objective, we considered seven simplifying assumptions:I.ME/CFS-diagnosed cases are a mix of apparent and genuine patients of the disease;II.The causal factor is only associated with genuine ME/CFS patients;III.Apparent cases are similar to healthy controls as far as the association with the causal factor is concerned;VI.The chance of an ME/CFS misdiagnosis is only dependent on the true clinical status of the cases and not on the confounding factors;V.The true association is independent of disease duration and disease triggers, among other factors occurring during the disease course;VI.Healthy controls were not misdiagnosed as such;VII.The value of the candidate causal factor can be determined perfectly in each individual.

The first assumption is simply the invocation of misdiagnosis in a strict sense (i.e., they are actually patients of another disease). The second assumption determines that there is a true association between the causal factor and ME/CFS. In the third assumption, we determine that the apparent ME/CFS cases share with healthy controls the same probability of the causal factor being present, $\theta_{0}$. The fourth and fifth assumptions simplify the determination of what a misdiagnosed case can be, linking it exclusively to the true/apparent category, thus, rejecting other potential disease-related factors that may influence the disease association. The sixth assumption aims at excluding the situation in which healthy controls could include undiagnosed genuine ME/CFS patients.

Note that the above assumptions are for mathematical convenience and represent the minimal set of conditions that enable the derivation of simple formulas for the probability of the causal factor being present in putative cases. As a consequence, the data simulation procedure is simplified. Additional assumptions can be invoked, but they would lead to a more-complex data simulation procedure. This is the case of also assuming that genuine cases are divided into several sub-types with different degrees of association with the causal factor. This situation, although more realistic, is beyond the scope of this paper due to its higher modeling complexity. On the other hand, the apparently different assumption in which misdiagnosis does not depend on the clinic and the clinicians who performed the diagnoses falls under the umbrella of the fourth assumption, where putative confounding factors would be given by a confounding factor referring to the participating clinics if applicable and another one referring to the clinicians.

Based on the above assumptions, the probability of the causal factor being present in ME/CFS-diagnosed cases can be expressed as follows
(1)$$\theta_{1}=\gamma \theta_{0}+\left(1-\gamma \right)\theta_{1}^{*}\;,$$
where γ is the probability of misdiagnosing an apparent case as a genuine one, and $\theta_{1}^{*}$ is the probability of the candidate causal factor being present in genuine ME/CFS cases. If misdiagnosis could be an observable outcome, the above 2×2 contingency table could be augmented as shown in Table A1 (Appendix A).

A more complex situation emerges from the previous scenario where the candidate causal factor cannot be determined perfectly in each individual. As a consequence, there is the possibility of having misdiagnosis together with misclassification of the causal factor. This is particularly relevant to serological studies that aim at investigating whether the presence of specific antibodies is associated with ME/CFS [23] or whether these antibodies can be used for disease diagnosis [24]. Note that the serological evaluation of a suspected case is not mandatory by consensual definitions of ME/CFS [1].

To model this new situation, the above assumption VII is replaced with two additional assumptions:VII.There are only two possible serological outcomes for each individual: seronegative or seropositive;VIII.The sensitivity and specificity of the serological classification are identical for all of the individuals.

The revised assumption VII excludes the situation where the serological classification can contemplate an indeterminate status due to the laboratory protocol [22] or the presence of multiple serological populations [25]. Similarly to assumption V for misdiagnosis, the new assumption VIII intends to disregard the effect of confounders (i.e., age or gender) and disease-related factors (i.e., disease duration or disease severity) on the performance of the serological classification.

Under the validity of assumptions I–VIII, the probability of the candidate causal factor being present in a ME/CFS-diagnosed patient can be extended to
(2)$$\theta_{1}=\pi_{se}\gamma \theta_{0}+\left(1-\pi_{sp}\right)\gamma \left(1-\theta_{0}\right)+\pi_{se}\left(1-\gamma \right)\theta_{1}^{*}+\left(1-\pi_{sp}\right)\left(1-\gamma \right)\left(1-\theta_{1}^{*}\right)\;,$$
where $\pi_{se}$ and $\pi_{sp}$ are the sensitivity and specificity for the serological classification, respectively; see Table A2 (Appendix A) for details. Note that when $\pi_{se}=\pi_{sp}=1$ (perfect serological testing), the above formula converts to Equation (Equation 1).

### 2.2. Simulation Study

To investigate the impact of the above misdiagnosis scenarios, we performed a comprehensive simulation study using the R statistical software, version 4.1.0 [26]. Individuals from each group were selected in accordance with the study’s sampling distribution, as shown in Equation (Equation 3) (Appendix B). We assumed the same sample size for ME/CFS patients and healthy controls (i.e., $n_{0}=n_{1}$, with $n_{0}$ and $n_{1}$ being the sample sizes for healthy controls and ME/CFS-diagnosed patients in each simulated scenario, respectively). We considered the following sample sizes per study group: 100, 250, 500, 1000, 2500, and 5000.

To parameterize the simulation study, we first specified the association between the candidate causal factor and genuine ME/CFS patients by the odds ratio (hereafter denoted as $\Delta_{T}$) and the probability of the presence of the causal candidate factor in healthy controls and apparent ME/CFS cases ($\theta_{0}$). We considered the true association (i.e., $\Delta_{T}$) between genuine ME/CFS cases and the causal factor to vary from weak to strong values (i.e., $\Delta_{T}\in \left\{1.25,1.5,2,3,5,10\right\}$). We also specified $\theta_{0}\in \left\{0.05,0.1,0.25,0.5\right\}$. If data comes from a genetic association study, $\theta_{0}$ could represent the minor allele frequency of a given single nucleotide variant in the healthy population. Note that, having $\theta_{0}$ and $\Delta_{T}$ fixed in the respective values, the value of $\theta_{1}^{*}$ can be estimated, as shown in Equation (Equation 4) (Appendix B). The misdiagnosis probability (or rate) γ was varied from 0 to 1 (all diagnosed individuals are genuine and apparent ME/CFS cases, respectively) with a lag of 0.01.

To simulate data from the second misdiagnosis scenario, we considered fixed parameters $\Delta_{T}=3$ and $\theta_{0}=0.25$. For parameters $\pi_{se}$ and $\pi_{sp}$, we considered all possible combinations of 0.80, 0.90, 0.925, 0.975, and 1.0, where $\pi_{se}=\pi_{sp}=1$ corresponded to the first scenario.

For each misdiagnosis scenario, parameter set, and sample size, we simulated 10,000 data sets to estimate the power of detecting an association under the presence of misdiagnosis. A detailed description of the simulation procedure can be found elsewhere [11,27]. In each data set, we rejected the presence of association if the *p*-value of Pearson’s $\chi^{2}$ test was greater than the usual 5% level of significance. For each parameter combination, the power (1−β) was estimated by the proportion of the simulated data sets in which an association was detected. To facilitate the understanding of the simulation results, we specified a target power of at least 80%.

### 2.3. Application to Two ME/CFS Studies

We also studied the impact of misdiagnosis on published data from a candidate gene association study and an immunological evaluation study. The first study recruited 201 healthy controls and 305 ME/CFS patients whose symptoms complied with the Canadian Consensus Criteria [21]. Five single-nucleotide polymorphisms (SNPs) were evaluated in all participants. The study found significant associations of rs2476601 and rs3087243 with ME/CFS whose onset was triggered by an acute infection.

The second study refers to serological data on 251 ME/CFS patients and 107 healthy controls from the UK ME/CFS Biobank [22]. These serological data referred to antibody positivity to each of six different herpesviruses: human cytomegalovirus (CMV), Epstein-Barr virus (EBV), herpes simplex virus 1 and 2 (HSV1 and HSV2), varicella-zoster virus (VZV), and human herpesvirus (HHV6). Antibody positivity per herpesvirus was previously determined by different lab protocols that did not provide any information about the specificity and sensitivity of the resulting serological classification.

In both studies, we estimated the power of detecting an association as a function of misdiagnosis probability, γ, using simulated data generated from the reported associations, as explained later.

## 3. Results

### 3.1. Simulation Study: Impact of ME/CFS Misdiagnosis

The power to detect an association with ME/CFS decreased with the misdiagnosis probability (Figure 1 and Figure 2). The maximum power was achieved when the diagnosed individuals were all genuine ME/CFS cases (γ=0). When the diagnosed individuals were all apparent ME/CFS cases (γ=1), the corresponding power matched the 5% significance level. This result was a direct consequence of assumption III, in which the misdiagnosed cases were considered identical to healthy controls as far as the association with the candidate causal factor was concerned.

As expected, the most optimistic scenarios were associated with $\Delta_{T}=5$ or 10 (i.e., strong associations between the candidate causal factor and ME/CFS). In these scenarios, one could find a maximum misdiagnosis probability for which the power of 80% was achieved (Table 2). For $\Delta_{T}=10$, a misdiagnosis probability of 0.53 was sufficient to ensure the desired power for sample sizes greater than or equal to 100 individuals per study group ($n_{i}\geq 100$), irrespective of $\theta_{0}$. This minimum probability was reduced to 0.24 for $\Delta_{T}=5$.

Similar optimistic scenarios were observed for sample sizes of 2500 and 5000 individuals per study group with the exception of the case of lowest $\Delta_{T}=1.25$. Combining these large sample sizes with strong associations between the candidate causal factor and the true ME/CFS cases, failing to achieve the target power only occurred when almost all the cases were misdiagnosed (with misdiagnosis probability greater than or equal to 0.88).

Unsurprisingly, the most pessimistic situations were related to $\Delta_{T}=1.25,1.5$, $n_{0}=n_{1}=100$, or a combination of the two. When $\Delta_{T}=1.25$, the sample size had to increase to 2500 or 5000 individuals per group in order to achieve the target power. Therefore, for this weak association, the chance of finding reproducible results was very low, even under the assumption of a perfect diagnosis. As a consequence, testing the “common disease, common variant hypothesis” in ME/CFS is likely to fail in future genetic associations. Finally, the case of $n_{0}=n_{1}=100$ was particularly problematic given that it was not possible to find any value misdiagnosis probability in which the desired power could be achieved for $\Delta_{T}\leq 2$ (Figure 1).

### 3.2. Simulation Study: Impact of ME/CFS Misdiagnosis and Misclassification on the Candidate Causal Factor

We then simulated the data of a hypothetical association study in which there were both imperfect diagnoses and misclassification of the candidate causal factor (Figure 2). This situation underpins any serological association study in ME/CFS, given the estimation of seropositivity of all individuals could be affected by the sensitivity and specificity associated with the classification rule used. At this point, it was clear that for values of $\Delta_{T}=1.25$, 1.5, and 2, the desired power was not often achieved for sample sizes smaller than 500 individuals per group in the case of perfect classification of the causal factor. Therefore, the additional assumption of imperfect classification of the candidate causal factor would make the previously estimated power even worse. Because of that, we only performed our simulation study on the more optimistic scenario in which $\Delta_{T}=3$ (Table 3).

### 3.3. Application to Data from Two ME/CFS Studies

We illustrated the problem of misdiagnosis in data from two ME/CFS studies [21,22]. We started with data from a candidate gene association study [21]. In this study, some genetic associations were only found to be significant when comparing healthy controls to ME/CFS patients with an infectious disease trigger onset (Table 4). The estimated allele-related odds ratios varied from 0.84 [95%CI=(0.56,1.27)] (rs1799724, *TNF*) to 1.63 [95%CI=(1.04,2.55)] (rs2476601, *PTPN22*). In our re-analysis, we investigated the impact of misdiagnosis if a replication study were conducted in a similar population. In line with the original study, no genotyping errors were assumed for the genetic data. The reported odds ratios were assumed to be the true ones for the population, and data were simulated with the same allele frequencies as reported in the original study.

Again, the estimated probability of detecting an association decreased with the misdiagnosis probability (Figure 3A). More importantly, when the misdiagnosis probability was low (γ<0.09), it was possible to achieve the minimum power of 80% for the allele association reported for rs3087243 in *CTLA4*. Therefore, the target power cannot be ensured for γ>0.09. For the remaining SNPs, the target power was never achieved, irrespective of the misdiagnosis probability. This is particularly problematic for rs2476601 in *PTPN22* whose association was reported to be significant at the 5% significance level. For this SNP, the misdiagnosis probability of approximately 0.10 had an estimated power of about 50%. This result implies that the chance of replicating the reported association was no better than flipping a coin.

The second study referred to putative associations of six herpes virus infections with ME/CFS using antibody positivity data [22]. In these data, all individuals were classified as seronegative or seropositive for each antibody used. Under the assumption of perfect serological classification and diagnosis, the associations of these serological data with severely affected ME/CFS patients ranged from 0.65 [95%CI=(0.21,1.97)] to 1.60 [95%CI=(0.83,3.09)] for EBV and HSV1, respectively (Table 5). In this study, no association was deemed significant at the usual significance level of 5%, according to the original study (*p*-values ≥ 0.16).

The original serological classification was based on a cut-off in the antibody levels determined by the 2σ rule; the cut-off is the mean plus twice the standard deviation of a known or hypothetical seronegative population. Under the assumption of a normal distribution for the seronegative population, the expected specificity of the serological specificity is approximately 0.975 [28]. We assumed this value for $\pi_{sp}$. For simplicity, we assumed $\pi_{se}=\pi_{sp}$. Again, we simulated data from this scenario as the original study and estimated the probability of detecting an association as a function of the misdiagnosis probability.

In this study, the minimum power of 80% could not be reached for any of the antibodies (Figure 3B). The best case was the antibody data related to HSV1. In this case, the maximal power was around 0.50 in the absence of misdiagnosis. This power dropped to 0.30 when γ=0.25. For the remaining cases, the power was almost less than 0.20. This could partially be explained by the fact that $\theta_{0}$ is higher than 0.93 for antibody data related to EBV, HHV6, and VZV.

## 4. Discussion

This study investigated the impact of misdiagnosis on the reproducibility of ME/CFS association studies. Our simulation study showed that strong associations with ME/CFS can be detected with reasonable power even under a non-negligible misdiagnosis rate. However, strong associations might not be the case of ME/CFS given the difficulty in finding a disease biomarker [29] and a clear genetic signature of the disease [30,31,32,33].

Studies with sample sizes larger than 500 individuals per study group are able to compensate for the reduction in power due to misdiagnosis alone. This minimum sample size increases when, besides misdiagnosis, there is also the possibility of not determining the presence of the causal factor perfectly. In general, large studies are becoming common in well-known and highly-funded diseases, such as cancer, cardiovascular diseases [34], and autoimmune disorders [35,36]. However, large ME/CFS studies are currently unfeasible due to limited funding and poor societal recognition of the disease [37]. This problem can be somehow minimized by using data from the United Kingdom ME/CFS Biobank that includes biological samples of more than 500 individuals [38]. Another solution is to conduct multi-centric studies [29]. Increasing sample size via data from self-reported ME/CFS cases (as performed in studies based on the UK Biobank) does not seem a viable solution because the chance of misdiagnosis is too high for obtaining reliable results. This problem is clearly illustrated in a Polish study where 1400 individuals were believed to be suffering from ME/CFS, but only 69 individuals actually complied with a consensual ME/CFS case definition [39].

Current serological association studies of ME/CFS neglect the possibility of misclassifying seropositive individuals. In addition, it is common to leave the sensitivity and specificity of the respective serological classification unreported. This research practice adds to the list of other factors that can contribute to the lack of reproducibility of ME/CFS serological studies [40]. Genetic association studies of ME/CFS also neglect the possibility of misclassifying the genotypes of the individuals. This neglect is reasonable in most studies given that genotype error rates are often below 1%, and rare genetic markers with higher genotype errors are typically excluded from the analysis [33,41,42].

Our results are based on the assumption that disease association is independent of possible confounding factors. This assumption seems appropriate for randomized clinical studies or studies based on the analysis of specific subgroups, such as only focusing on adult women with an infection at the disease onset. However, it is also known that age, gender, and exposure to a given infectious agent can affect the results [43,44]. Therefore, the assumption might not be true in general.

Our results are also based on the assumption that the controls are indeed healthy. Interestingly, ME/CFS patients and some healthy controls might have the same symptoms profile and similar levels of fatigue [11,45]. More importantly, the use of self-reported healthy controls [44,46] or control samples from existing blood banks [47,48,49] are also common practices in ME/CFS research. According to these research practices, a more realistic assumption is to divide healthy controls into genuine and apparent controls. However, we anticipate that the statistical power to detect a putative disease association is further reduced in this more general scenario. To avoid this scenario, a thorough clinical assessment should also be performed in putative healthy controls.

This study was framed in terms of ME/CFS misdiagnosis in a strict sense. However, from a modelling standpoint, this framing is mathematically equivalent to the situation where ME/CFS-diagnosed cases can be partitioned into two subgroups of genuine patients but with distinct pathological mechanisms and where the association is only present in one of these subgroups. Therefore, our results are directly applicable to this alternative situation but with caution. As alluded to in the introduction, ME/CFS might not be one but several diseases under the same umbrella term, as suggested by genomic data [50,51]. Having said that, a more realistic situation is to have multiple subgroups with different degrees of association with the potential causal factor. Therefore, there is a need to extend our simulation study to this situation.

In conclusion, current case-control association studies of ME/CFS seem to have limited power to mitigate the effect of misdiagnosis in the detection of putative disease associations. A sample size of 500 or 1000 individuals per study group is a minimal requirement to detect mild-to-moderate associations with a high power under the assumption of misdiagnosis. These sample sizes are attainable from multi-centric studies; these studies require extensive collaboration among ME/CFS researchers. Under the impossibility of increasing sample size, research efforts should be made towards reducing the rate of strict misdiagnosis. This can be achieved by following existing recommendations for research reports of ME/CFS, such as reporting the screening laboratory tests and the cut-off values for exclusion [52]. It can also be achieved by the continued search for alternative diagnoses and co-morbidities [53]. In the end, a better understanding of multiple disease pathways leading to ME/CFS leads to better diagnoses, and, therefore, one should ultimately aim to study homogeneous cohorts of patients where the chance of strict misdiagnosis is reduced.

## Figures and Tables

**Figure 1 diagnostics-13-00531-f001:**
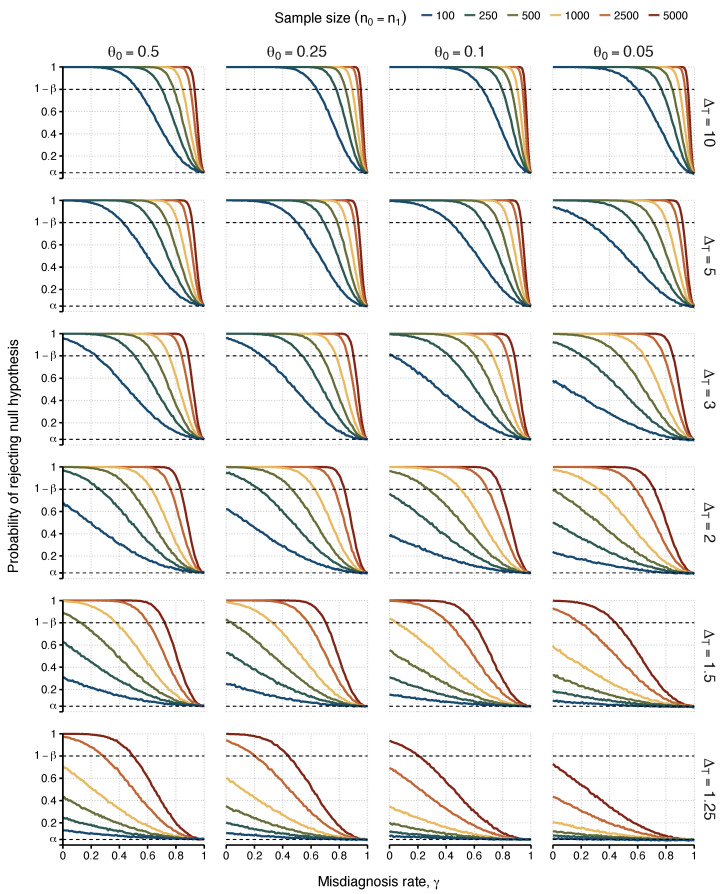
Probabilities of detecting an association (i.e., rejecting $H_{0}$) as a function of the misdiagnosis rate. Each column represents the values attributed to the risk allele frequency found in matched healthy controls and false positive ME/CFS cases ($\theta_{0}\in \left\{0.05,0.1,0.25,0.5\right\}$). Each row varies the true odds ratio for the association between risk allele frequency assessed between true positive cases and healthy controls ($\Delta_{T}\in \left\{1.25,1.5,2,3,5,10\right\}$). Power was estimated for different sample sizes of 100, 250, 500, 1000, 2500, and 5000 ($n_{0}=n_{1}$), represented by lines with different colours in each scenario. The upper dashed line indicates the target power of 80% (i.e., 1−β=0.80). The lower dashed line indicates the 5% significance level.

**Figure 2 diagnostics-13-00531-f002:**
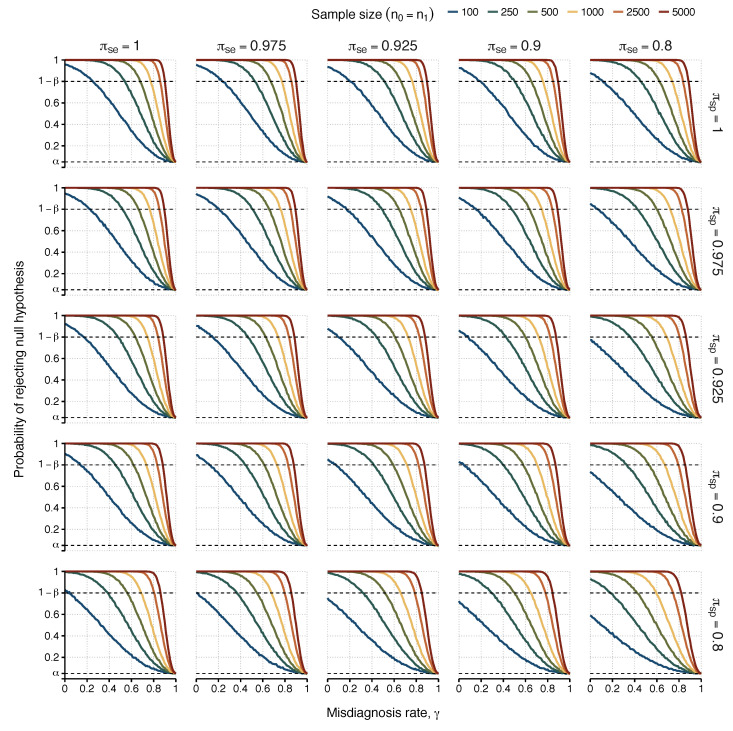
Probabilities of detecting an association (i.e., rejecting $H_{0}$) as a function of the misdiagnosis rate. Each scenario represents simulated results with a different combination of sensitivity ($\pi_{se}$) and specificity $\pi_{sp}$ for the serological test for columns and rows, respectively. Power was estimated for different sample sizes of 100, 250, 500, 1000, 2500, and 5000 ($n_{0}=n_{1}$), represented by lines with different colours in each scenario, with the probability of exposure in healthy controls fixed as $\theta_{0}=0.25$ and true odds ratio $\Delta_{T}=3$. The upper dashed line indicates the target power of 80% (i.e., 1−β=0.80). The lower dashed line indicates the 5% significance level.

**Figure 3 diagnostics-13-00531-f003:**
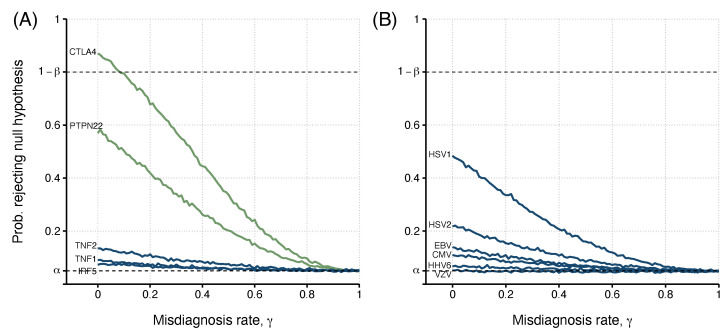
The relationship between the misdiagnosis probability (or rate) and the probability of detecting an association (i.e., rejecting the $H_{0}$) estimated from simulated data from two previously published studies: (**A**). Data from five different SNPs (genes *PTPN22*, *CTLA4*, *TNF* (TNF1 - rs1799724 and TNF2 - rs1800629), and *IRF5*); (**B**). Data of antibody positivity related to six human herpesviruses (CMV, EBV, HSV1 and HSV2, VZV, and HHV6). For each study, risk allele frequencies or the probability of exposure and true odds ratio were determined by Steiner et al. [21] ($n_{0}=201$; $n_{1}=305$) and Cliff et al. [22] ($n_{0}=107$; $n_{1}=251$; $\pi_{se}=\pi_{sp}=0.975$), with determined values shown in Table 4 and Table 5, respectively. Green lines indicate candidate risk factors where a significant association with the disease was found in the original study. Blue lines show non-significant ME/CFS risk factors. The upper dashed line indicates the target power, where the probability of rejecting the null hypothesis is 1−β=0.80. The lower dashed line indicates the significance level used, α=0.05.

**Table 1 diagnostics-13-00531-t001:** Two-way contingency table of a typical case-control study where $\theta_{0}$ and $\theta_{1}$ are the probabilities of the candidate causal factor being present in healthy controls and ME/CFS-diagnosed cases, respectively.

Causal Factor	Controls	ME/CFS-Diagnosed Cases
Present	$\theta_{0}$	$\theta_{1}$
Absent	$1-\theta_{0}$	$1-\theta_{1}$

**Table 2 diagnostics-13-00531-t002:** Maximum values of misdiagnosis probability γ that ensure the minimum power of 80% to detect a genuine association $\Delta_{T}$ as a function of $\theta_{0}$, and sample size *n* per group. Cells with no value indicate that the minimum power could not be reached in the respective parameter combination.

	$\theta_{0}$	0.05	0.1	0.25	0.5	*n* (per Group)
$\Delta_{T}$						
10		0.59	0.65	0.64	0.53	100
5		0.24	0.43	0.50	0.42	
3		−	0.02	0.25	0.23	
2		−	−	−	−	
1.5		−	−	−	−	
1.25		−	−	−	−	
10		0.77	0.79	0.77	0.70	250
5		0.56	0.66	0.69	0.63	
3		0.20	0.41	0.53	0.50	
2		−	−	0.23	0.26	
1.5		−	−	−	−	
1.25		−	−	−	−	
10		0.84	0.86	0.84	0.78	500
5		0.70	0.76	0.78	0.73	
3		0.47	0.60	0.67	0.65	
2		−	0.27	0.46	0.47	
1.5		−	−	0.04	0.13	
1.25		−	−	−	−	
10		0.89	0.90	0.89	0.84	1000
5		0.80	0.84	0.85	0.81	
3		0.64	0.72	0.77	0.75	
2		0.32	0.50	0.62	0.62	
1.5		−	0.05	0.32	0.38	
1.25		−	−	−	−	
10		0.93	0.94	0.93	0.90	2500
5		0.88	0.90	0.90	0.88	
3		0.78	0.83	0.85	0.84	
2		0.58	0.69	0.76	0.76	
1.5		0.18	0.42	0.58	0.59	
1.25		−	−	0.20	0.28	
10		0.95	0.95	0.95	0.93	5000
5		0.91	0.93	0.93	0.91	
3		0.84	0.88	0.90	0.88	
2		0.71	0.78	0.83	0.83	
1.5		0.44	0.59	0.70	0.72	
1.25		−	0.20	0.44	0.49	

**Table 3 diagnostics-13-00531-t003:** Maximum values of misdiagnosis probability γ that still ensure a power of rejecting the null hypotheses of at least 80% for $\Delta_{T}=3$ and $\theta_{0}=0.25$, where $\pi_{se}$ and $\pi_{sp}$ represent sensitivity and specificity associated with the classification of the candidate, respectively. See Table 2 for more information.

	$\pi_{\mathrm{se}}$	1	0.975	0.925	0.9	0.8	*n* (per Group)
$\pi_{\mathrm{sp}}$							
1		0.25	0.23	0.20	0.19	0.11	
0.975		0.22	0.20	0.17	0.15	0.06	
0.925		0.17	0.14	0.09	0.08	−	100
0.9		0.13	0.11	0.07	0.04	−	
0.8		0.03	−	−	−	−	
1		0.53	0.52	0.51	0.50	0.45	
0.975		0.51	0.50	0.48	0.47	0.42	
0.925		0.47	0.46	0.43	0.42	0.36	250
0.9		0.45	0.43	0.41	0.39	0.32	
0.8		0.38	0.36	0.31	0.29	0.18	
1		0.67	0.67	0.66	0.65	0.62	
0.975		0.66	0.65	0.64	0.63	0.59	
0.925		0.63	0.62	0.60	0.59	0.55	500
0.9		0.61	0.61	0.59	0.57	0.52	
0.8		0.56	0.54	0.51	0.50	0.42	
1		0.77	0.77	0.76	0.75	0.73	
0.975		0.76	0.75	0.74	0.74	0.72	
0.925		0.74	0.73	0.72	0.71	0.68	1000
0.9		0.73	0.72	0.71	0.70	0.67	
0.8		0.68	0.67	0.65	0.64	0.59	
1		0.85	0.85	0.85	0.84	0.83	
0.975		0.85	0.85	0.84	0.84	0.82	
0.925		0.84	0.83	0.82	0.82	0.80	2500
0.9		0.83	0.82	0.81	0.81	0.79	
0.8		0.80	0.79	0.78	0.78	0.74	
1		0.90	0.90	0.89	0.89	0.88	
0.975		0.89	0.89	0.89	0.88	0.87	
0.925		0.88	0.88	0.87	0.87	0.86	5000
0.9		0.88	0.87	0.87	0.87	0.85	
0.8		0.86	0.85	0.84	0.84	0.81	

**Table 4 diagnostics-13-00531-t004:** Reported associations of a candidate gene association study [21] where ${\hat{\theta }}_{0}$ represents the frequencies of the non-reference allele for healthy controls and ${\hat{\Delta }}_{T}$ is the odds ratio of these allele frequencies when comparing ME/CFS patients with an infectious disease trigger to healthy controls. *p*-values are associated with the Pearson’s $\chi^{2}$ test for two-way contingency tables.

SNP	Gene	${\hat{\theta }}_{0}$	${\hat{\Delta }}_{T}$	95% CI ( ${\hat{\Delta }}_{T}$)	*p*-Value
rs3087243	*CTLA4*	0.56	1.54	(1.17, 2.03)	0.002
rs2476601	*PTPN22*	0.08	1.63	(1.04, 2.55)	0.033
rs1799724	*TNF*	0.13	0.84	(0.56, 1.27)	0.409
rs1800629	*TNF*	0.16	0.89	(0.61, 1.30)	0.551
rs3807306	*IRF5*	0.51	0.94	(0.72, 1.22)	0.637

**Table 5 diagnostics-13-00531-t005:** Summary of serological findings from [22], where ${\hat{\theta }}_{0}$ represents the seroprevalence of healthy controls, and ${\hat{\Delta }}_{T}$ refers to the odds ratio for being seropositive when comparing severely affected ME/CFS patients to healthy controls. The 95% CI (${\hat{\Delta }}_{T}$) and *p*-values are associated with the Pearson’s $\chi^{2}$ test for two-way contingency tables.

Herpes Virus	${\hat{\theta }}_{0}$	${\hat{\Delta }}_{T}$	95% CI (${\hat{\Delta }}_{T}$)	*p*-Value
HSV1	0.42	1.60	(0.83, 3.09)	0.163
HSV2	0.34	1.36	(0.69, 2.66)	0.377
EBV	0.93	0.65	(0.21, 1.97)	0.442
CMV	0.37	0.84	(0.42, 1.67)	0.613
VZV	0.97	0.75	(0.12, 4.63)	0.757
HHV6	0.95	1.27	(0.24, 6.79)	0.776

## Data Availability

The source codes and all the simulated data can be downloaded freely from https://github.com/jtmalato/misclassification-simulations (accessed on 10 October 2022).

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
