# Peer review of "Impact of Misdiagnosis in Case-Control Studies of Myalgic Encephalomyelitis/Chronic Fatigue Syndrome"

_diagnostics, 2023, doi:10.3390/diagnostics13030531_

Round 1

Reviewer 1 Report

The authors present a statistical model for calculating the number of participants needed for identifying risk factors for ME/CFS. This study emphasises the need for large study populations to get enough power. This is an important question in clinical research and the authors conclusion is worth to consider.

There are, however, some problems that should be taken in consideration

Misdiagnosis – what is that in ME? ME -diagnosis is put according to criteria. If the criteria are fulfilled the patient has the diagnosis! (according to the specific criteria). Misdiagnosis would then be putting the diagnosis although the criteria are not fulfilled. The problem is that there are many different published sets of criteria. The differences could exampled by the CDC-criteria (Fukuda et al 1994) where post exertional malaise with prolonged recovery period is not a compulsory symptom in contrast with the Canada-criteria or SEID (IOM). This means that there are different ME-diagnoses according to which criteria are used.

The pathophysiology of ME has not been set. This is probably due to that ME is not ONE disease. Several pathophysiological mechanisms have been proposed, e.g. leaking gut, autoantibodies (e.g. adrenergic and cholinergic receptors), oxidative stress, persisting virus, genetic aberrations, severe longstanding stress etc

Looking for biomarkers in a disease with perhaps ten different pathophysiological mechanisms probably need very large samples (as this paper clearly demonstrates) – but is this really relevant? ME-research should probably benefit from studying subpopulations with a defined pathophysiologic mechanism.

Those to problems must be discussed.

There is a long discussion, could be shorter, and no conclusion

I general the manuscript should be shortened

Author Response

The authors present a statistical model for calculating the number of participants needed for identifying risk factors for ME/CFS. This study emphasises the need for large study populations to get enough power. This is an important question in clinical research and the authors conclusion is worth to consider.

There are, however, some problems that should be taken in consideration

Misdiagnosis – what is that in ME? ME -diagnosis is put according to criteria. If the criteria are fulfilled the patient has the diagnosis! (according to the specific criteria). Misdiagnosis would then be putting the diagnosis although the criteria are not fulfilled. The problem is that there are many different published sets of criteria. The differences could exampled by the CDC-criteria (Fukuda et al 1994) where post exertional malaise with prolonged recovery period is not a compulsory symptom in contrast with the Canada-criteria or SEID (IOM). This means that there are different ME-diagnoses according to which criteria are used.

Response: Thank you for bringing up this key point of our paper. We believe there are two senses of misdiagnosis in the context of ME/CFS. Relative misdiagnosis refers to your point where the same suspected case might have a different diagnostic outcome using one or another definition. This type of misdiagnosis is beyond the scope of this paper, because it involves more a discussion about different case definitions. The reader can find good discussions on this topic in Brurberg et al (ref. 19) and Lim and Son (new ref. 20). In this paper, we tackled misdiagnosis in a strict sense where an ME/CFS-diagnosed individual is actually a genuine patient of another disease. This type of misdiagnosis can arise from: (i) random fluctuations in the pathological process of the true disease (e.g., remitting/relapsing multiple sclerosis); (ii) not running all clinical tests to exclude all known diseases; (iii) ambiguity in the exclusionary criteria which leave clinicians/researchers guessing which conditions/comorbidities to exclude (new ref. 17). We have now revised the abstract and explored these two concepts of misdiagnosis in the second and third paragraphs of the Introduction (lines 24-46). 

The pathophysiology of ME has not been set. This is probably due to that ME is not ONE disease. Several pathophysiological mechanisms have been proposed, e.g. leaking gut, autoantibodies (e.g. adrenergic and cholinergic receptors), oxidative stress, persisting virus, genetic aberrations, severe longstanding stress etc

Response: Thank you for the comment. We have now alluded to this point at the first paragraph of the introduction (lines 20-22) and in the Discussion (lines 312-316).   

Looking for biomarkers in a disease with perhaps ten different pathophysiological mechanisms probably need very large samples (as this paper clearly demonstrates) – but is this really relevant? ME-research should probably benefit from studying subpopulations with a defined pathophysiologic mechanism.

Response: Thank you for the comment. We have now included this point in the discussion (lines 326-329). 

Those to problems must be discussed.

There is a long discussion, could be shorter, and no conclusion

Response: Thank you for the suggestion. We have now shortened the Discussion by removing all tangent points. We have also added a concluding paragraph at the end of the manuscript (lines 317-329).

I general the manuscript should be shortened

Response: Thank you for the suggestion. We have now shortened all Sections including the abstract. The main text is now 12-page long in contrast with 15 pages of the original submission; the number of lines decreased from 460 to 329; the number of cited papers decreased from 67 to 53. We essentially removed all non-essential points and revised the Written-English to be simpler and more concise. We believe the revised version is more straightforward to anyone to read.   

Reviewer 2 Report

The paper could be improved as follows:

Introduction

     2nd paragraph, 2nd sentence: Would change "To make the disease diagnosis even more cumbersome," to: Unfortunately, 

     2nd paragraph, end: I would add a sentence something like: The IOM  criteria attempted to improve upon earlier definitions by focusing on the key and most debilitating symptoms, although they lack exclusionary criteria.  

     3rd paragraph, 4th sentence.  There is much less overlap between cases and controls when frequency and severity of symptoms is taken into account (Jason, Fatigue. 2014 January 1; 2(1): 40–56. doi:10.1080/21641846.2013.862993.)   This will also need to be addressed in the 5th paragraph of the Discussion.  

     3rd paragraph: I would delete the last sentence.  

     4th paragraph: I would change "scenario" to: study, and "under different sample sizes" to: using different sample sizes.  

     4th paragraph,  2nd to last sentence: The authors need to justify why they felt serological studies were warranted, as the Fukuda criteria specifically state that there is no reason to perform them.  The discussion at the top of page 4 will also have to address this issue.

Statistical Methodology

     It is not clear to me that the 7 assumptions presented are exhaustive, so why were these 7 chosen?  I would also change "in what" in assumption III to: as far as.

Discussion

     2nd paragraph: If the diagnosis of ME/CFS is self-reported, that is a serious limitation, and it is hard to see how increasing the sample size can help.  

References

     Jason, Fatigue. 2014 January 1; 2(1): 40–56. doi:10.1080/21641846.2013.862993 needs to be added (see above).

     Reference 11 is not peer-reviewed.  

Author Response

The paper could be improved as follows:

Introduction

2nd paragraph, 2nd sentence: Would change "To make the disease diagnosis even more cumbersome," to: Unfortunately, 

Response:

Thank you for the suggestion. Due to the request of the other reviewer to shortening the paper, the Introduction was fully revised and this sentence was removed.  

2nd paragraph, end: I would add a sentence something like: The IOM  criteria attempted to improve upon earlier definitions by focusing on the key and most debilitating symptoms, although they lack exclusionary criteria.  

Response:

Thank you for the suggestion. As mentioned above, the second paragraph was fully revised and this sentence was also removed. 

3rd paragraph, 4th sentence.  There is much less overlap between cases and controls when frequency and severity of symptoms is taken into account (Jason, Fatigue. 2014 January 1; 2(1): 40–56. doi:10.1080/21641846.2013.862993.)   This will also need to be addressed in the 5th paragraph of the Discussion.  

Response: Thank you for highlighting this reference. We have made the changes in lines 33-35.

3rd paragraph: I would delete the last sentence.  

Response: Thank you for the suggestion. Due to the request of the other reviewer to shortening the paper, this paragraph was fully revised and the referred sentence removed. 

4th paragraph: I would change "scenario" to: study, and "under different sample sizes" to: using different sample sizes.  

Response: Thank you for the suggestions. We have now revised the full paragraph (lines 47-53).  

4th paragraph, 2nd to last sentence: The authors need to justify why they felt serological studies were warranted, as the Fukuda criteria specifically state that there is no reason to perform them.  The discussion at the top of page 4 will also have to address this issue.

Response: Thank you for raising this point. Our motivation to evaluate in the impact of specificity/sensitivity on serological stemmed from the fact that there are plenty of case-control studies aiming at understanding the role of specific (autoimmune, antiviral, among others) antibodies in the pathogenesis of ME/CFS. There is also an interest in discovering antibodies that could serve as biomarker for disease diagnosis. We have now clarified this motivation in lines 51-52 and 105-111.       

Statistical Methodology

It is not clear to me that the 7 assumptions presented are exhaustive, so why were these 7 chosen?  I would also change "in what" in assumption III to: as far as.

Response: Thank you for raising this key point about. These 7 assumptions are for mathematical convenience and represent the minimal set of conditions that enable to write simple probability formulas and simple simulation procedure for generating data. Other assumptions could be invoked in our study, but they would lead to a more complex simulation scenario. For example, we could have assumed that true cases are divided into multiple subtypes. This assumption, although realistic, is more complex to model. An apparent different assumption could be that the misdiagnosis is not dependent on the clinic or clinicians who conduct the diagnosis. However, this assumption is mathematically equivalent to the assumption IV which states that ME/CFS diagnosis is not dependent on the confounders. In that case, the confounders would be the covariates referring to the participating clinics (if applicable) and clinicians who performed the diagnosis. We have now clarified this point in lines 89-100. I also changed “in what” to “as far as” (line 71).   

Discussion

2nd paragraph: If the diagnosis of ME/CFS is self-reported, that is a serious limitation, and it is hard to see how increasing the sample size can help.

Response: Thank you for bringing up this point of the discussion. We fully agree that, if ME/CFS is self-reported, increasing sample size does not help, because the misdiagnosis probability might be extremely high. This has been demonstrated in Polish study in which 1400 individuals were thought to have ME/CFS, but only 69 complied with the Canadian Consensus Criteria (ref. 38). The changes made to the text are available in lines 278-283.   

References

Jason, Fatigue. 2014 January 1; 2(1): 40–56. doi:10.1080/21641846.2013.862993 needs to be added (see above).

Response: We have now cited this paper (new ref. 12).

Reference 11 is not peer-reviewed.  

Response: Thank you for pointing it out. Actually, ref. 11 is a preprint of a paper that is already published. We updated this reference accordingly.